# AIGCs Confuse AI Too: Investigating and Explaining Synthetic Image-induced Hallucinations in Large Vision-Language Models

Yifei Gao
Beijing Jiaotong University
Beijing, China
yifeigao@bjtu.edu.cn

Jiaqi Wang
Beijing Jiaotong University
Beijing, China
jiaqiw@bjtu.edu.cn

Zhiyu Lin
Beijing Jiaotong University
Beijing, China
zyllin@bjtu.edu.cn

Jitao Sang*
Beijing Jiaotong University
Beijing, China
Peng Cheng Lab
Shenzhen, China
jtsang@bjtu.edu.cn

## ABSTRACT

The evolution of Artificial Intelligence Generated Contents (AIGCs) is advancing towards higher quality. The growing interactions with AIGCs present a new challenge to the data-driven AI community: While AI-generated contents have played a crucial role in a wide range of AI models, the potential hidden risks they introduce have not been thoroughly examined. Beyond human-oriented forgery detection, AI-generated content poses potential issues for AI models originally designed to process natural data. In this study, we underscore the exacerbated hallucination phenomena in Large Vision-Language Models (LVLMs) caused by AI-synthetic images. Remarkably, our findings shed light on a consistent AIGC **hallucination bias**: the object hallucinations induced by synthetic images are characterized by a greater quantity and a more uniform position distribution, even these synthetic images do not manifest unrealistic or additional relevant visual features compared to natural images. Moreover, our investigations on Q-former and Linear projector reveal that synthetic images may present token deviations after visual projection, thereby amplifying the hallucination bias.

## CCS CONCEPTS

• **Computing methodologies → Artificial intelligence**.

## KEYWORDS

Artificial Intelligence Generated Contents, Large Vision Language Models, Objects Hallucination

---

*Corresponding authors

**ACM Reference Format:**
Yifei Gao, Jiaqi Wang, Zhiyu Lin, and Jitao Sang. 2024. AIGCs Confuse AI Too: Investigating and Explaining Synthetic Image-induced Hallucinations in Large Vision-Language Models. In *Proceedings of the 32nd ACM International Conference on Multimedia (MM '24), October 28-November 1, 2024, Melbourne, VIC, AustraliaProceedings of the 32nd ACM International Conference on Multimedia (MM'24), October 28-November 1, 2024, Melbourne, Australia.* ACM, New York, NY, USA, 9 pages. https://doi.org/10.1145/3664647.3681467

## 1 INTRODUCTION

With the rapid evolution of generative model techniques, Artificial Intelligence Generated Contents (AIGCs) have ushered in a new era of prosperity [22, 26]. AIGCs are no longer mere outputs of generative models; rather, they encompass the information generated during human-model or model-model interactions [3]. This result in a large amount of synthetic content rapidly flooding into the Internet, and people may have interacted with synthetic content unconsciously.

Nevertheless, the pervasive AI-generated content may give rise to several challenges. The first challenge to gain widespread attention is forgery detection [21]. This field aims to assist humans in distinguishing between natural and synthetic content and is considered a crucial aspect in AI safety. Particularly, a recent study [18] has indicated that the recognition of synthetic images has resulted in an approximate 40% human error rate, solidifying the fact that humans are easily confused by AIGCs. Another under-examined challenge is the impact of AI-generated content on AI models themselves. As synthetic data plays a more common role in the training and reasoning process [28], the hidden risks of AIGCs to the AI models are urgent yet largely unexplored. Taking Figure 1 as an example, synthetic images with identical semantics are more likely to induce hallucinations in LVLMs than natural images. This is largely because these models' training data, architecture, and training processes are inherently designed for natural data. Applying models trained on natural data to synthetic datasets without complications could lead to unexpected outcomes.

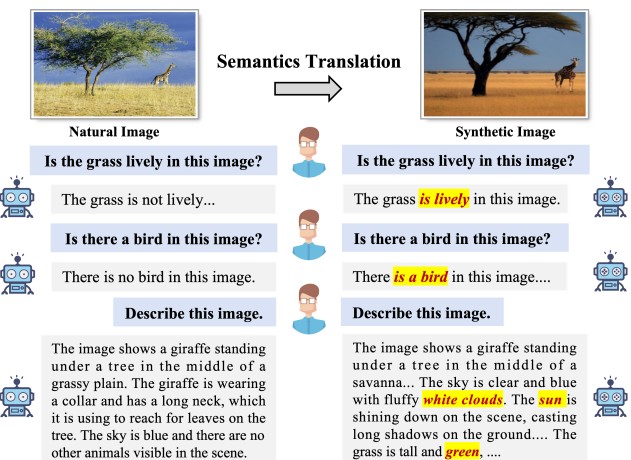

**Figure 1: A hallucination example on both synthetic (right) and natural images (left), where the highlighted fonts indicate the hallucinated content. Evaluation results across various vision-language tasks, such as semantic descriptions and factual judgments, consistently illustrate the existence of a synthetic image-induced hallucination bias.**

We are motivated to explore the topics of how synthetic data may impact AI models. In particular, this study focuses on the hallucination issues that synthetic images may cause in Large Vision-Language Models (LVLMs). Before addressing the core research question, we first establish a synthetic image-involved hallucination evaluation environment for LVLMs. Many current generative models adopt the Text-to-Image synthesis approach [22, 30]. However, the generation process leads to two primary issues: 1) semantic distortion [31], where synthetic images lack authenticity (e.g., the finger problem [18]) and; 2) semantic ambiguity [19], where synthetic images lack consistency and struggle to respond to text prompts. Given the absence of an available synthetic image dataset for hallucination evaluation, mitigating the impact of the aforementioned issues is necessary. To this end, we introduce a Semantics Translation (ST) method, which begins with a natural image and employs 1) caption generation and revision, and 2) semantic filtering strategies to control the authenticity and consistency of the synthetic image, ensuring that the evaluation is not affected by the quality of the synthetic image.

We translate two widely used hallucination evaluation datasets: POPE [13] and AMBER [23], and delve into the hallucinations induced by synthetic images. We also compare them with the corresponding natural images. Surprisingly, our findings indicate that LVLMs have a bias towards synthetic images, as shown in Figure 1. We refer to this phenomenon as synthetic image-induced hallucination bias (shorten as hallucination bias). Our further experiments reveal that the hallucination bias mainly exhibits 1) a greater quantity and 2) a more uniform position distribution of hallucinated content. Particularly, these phenomena are corroborated across different LVLMs and evaluation datasets. In other words, these LVLMs appear to adopt some inherent non-semantic shortcuts in synthetic images, which lead to a continuous impact on the extrapolation process. Then, we are committed to further studying

how synthetic images confuse LVLMs. Drawing inspiration from the visual projection process of LVLMs, we examine two prevalent visual projection modules: Q-former and Linear. Specifically, our investigations shed light on the fact that 1) turning off the Q-former projection or 2) deepening the layers of the Linear projection can 1) effectively mitigate the token deviation of synthetic images and 2) narrow the synthetic image-induced hallucination bias. That is to say, LVLMs tuned in this way may generate less hallucinated content in response to synthetic images.

Our core contributions are as follows: (1) In the context of the rapid development of AIGC, we explore the impact of synthetic images on the hallucination problem of LVLMs for the first time. To achieve this, we introduce a semantics translation method to establish a synthetic image-involved hallucination evaluation environment. (2) Extensive experiments uncover the synthetic image-induced hallucination bias of LVLMs, mainly manifesting as (i) a greater quantity, and (ii) a more uniform position distribution. (3) We provide an in-depth analysis on the synthetic image-induced hallucination bias from the perspective of visual-text alignment. Experimental results reveal that the current design of the visual projection module may cause the token deviation of synthetic images, thus resulting in the hallucination bias.

## 2 RELATED WORKS

**AIGC and its Challenges**: Recent advances in Artificial Intelligence Generated Contents (AIGCs) have profoundly transformed the approach to contents generation, and offered numerous benefits for humans in various aspects of life and work [25]. For instance, generative models such as Stable Diffusion [19] or ChatGPT-4 [1] can generate high-quality image or text information by adhering to the textual descriptions prompted by users. However, as AIGCs are progressively introduced into the online world and applied to society, it's essential to take note of certain potential hidden risks they may introduce. The first challenge to gain widespread attention is fake detection, which has been seen as the boundary of AI safety [21]. Another prevailing challenge stems from the insufficient examination on the impact of AI-generated content on AI models themselves. Specifically, certain researches have identified multifaceted risks associated with AIGCs, including issues related to security, bias, and privacy breaches [3, 4, 8, 24]. Furthermore, recent studies [6, 27] have demonstrated that synthetic data can introduce source bias in both text and cross-modal retrieval for web search, leading to an elevated ranking of AI-generated content (AIGCs). Therefore, within the context of the widespread application of AIGC, it is crucial to thoroughly investigate the potential risks it may pose.

**Hallucinations of LVLMs**: Large Vision Language Models [2, 16, 29, 34] (LVLMs) are regarded as a natural extension of Large Language Models [32] (LLMs). Through rich visual-text instructions-tuning, LVLMs have demonstrated remarkable progress in tackling complex multi-modal tasks, such as Visual Grounding and Visual Question Answering (VQA) [2, 16, 29, 34]. However, LVLMs are also plagued by the hallucination problem [10, 13, 17, 23], in which the model either 1) depicts inaccurate objects or 2) entirely fabricates content from associated images. These phenomena represent a major bottleneck in their deployment and thus limit their practicality in many scenarios [5, 33]. Recent research has delved into

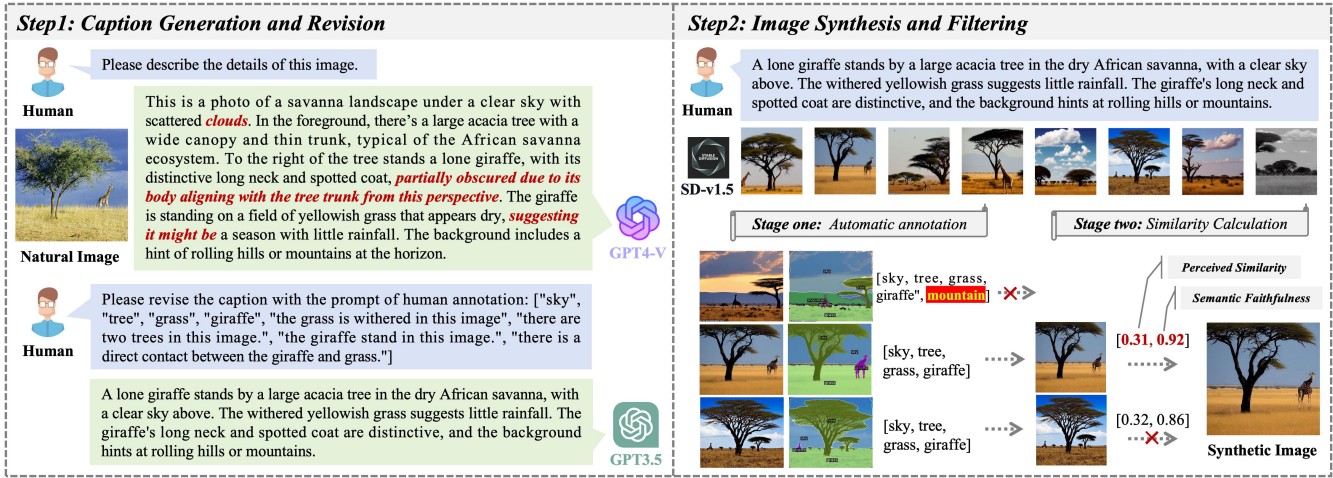

**Figure 2: The pipeline of semantics translation method. On the left side, we introduce caption generation and revision method to synthesize a correct description of the given natural image. *Red* represents the redundant or incorrect information within the initial caption. On the right side, we utilize image synthesis and filtering strategy to sample the final synthetic image, ensuring a strict correspondence to the revised caption and the input natural image. highlighted represents the redundant object in image synthesis process. The final synthetic image satisfies the criteria of authenticity and consistency.**

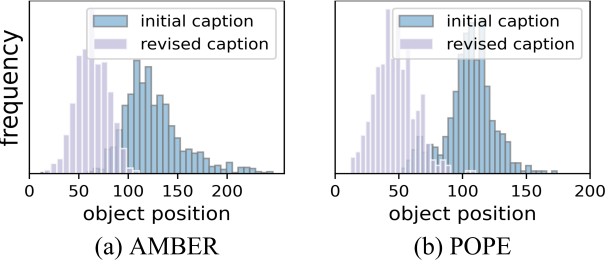

**Figure 3: The comparison of key object positions before and after the caption revision. Taking Stable Diffusion v1.5 as an example, where the accepted character limit is 77, the distribution of key objects in the revised caption generally satisfies the limits.**

hallucination problems from the perspective of evaluation and mitigation [17]. In terms of evaluation, POPE [13] defines the hallucination problem as a binary classification task, aiming to explore the model's perceptual ability with respect to specific objects present in the image. Meanwhile, AMBER [23] has contributed to the most comprehensive hallucination evaluation dataset by extending the binary evaluation approach of POPE and introducing generative task evaluation. The mitigation of hallucination generally involves three aspects: (1) Pre-processing, commonly achieved by providing high-quality image-text pairs and well-designed instruction-tuning [14]; (2) In-processing, where the model is enhanced by strengthening visual-textual feature learning [9]; (3) Post-hoc processing, known for its superior scalability, usually alleviates model hallucinations in the decoding stage [11].

## 3 SEMANTICS TRANSLATION

We focus on ensuring accurate hallucination evaluation on synthetic images, with the precondition of excluding the quality influence.

Specifically, a synthetic image should meet: 1) authenticity, where synthetic semantics should align with human cognition; and 2) consistency, where the synthetic image should accurately respond to the text prompt. In this section, we introduce a semantics translation method to synthesize high-quality images. As shown in Figure 2, the synthesis process of semantics translation method is constrained under a natural image supervision through the following two steps: 1) Caption Generation and Revision, transforming visual semantics into detailed textual semantics(Section 3.1); and 2) Image Synthesis and Filtering, involving (i) image over-sampling and (ii) image filtering based on similarity (Section 3.2).

### 3.1 Caption Generation and Revision

In this subsection, we translate the visual semantics of the given natural image into the textual semantics. As shown in Figure 2 (left), we employ GPT-4V(ision) to capture the key semantics. To maintain accurate textual semantics, we revise the extracted information through GPT-3.5.

**Caption Generation**: In order to ensure that the generative model receives text prompts closely aligned with the semantics of the given natural image, we employ GPT-4V to obtain coarse-grained captions. However, two notable issues persist in the generated captions: 1) redundant or missing information, where the generated semantics do not exist or fail to include the object that should be presented in the image; and 2) excessive caption length, where the generated captions often exceed the word limit accepted by most generative models. This results in the loss of semantic information beyond the word limit, thereby disrupting the consistency of semantics.

**Caption Revision**: To mitigate the aforementioned issues, the generated caption are revised by GPT-3.5. Specifically, we provide manual annotations to assist GPT-3.5 in comprehending and revising the

**Table 1: The overall evaluation results of POPE on both synthetic and natural images. Δ indicates the hallucination gap between natural and synthetic images. We use Δ represent the synthetic image-induced hallucination bias.**

| Model | Image | Random | | | Popular | | | Adversarial | | |
|---|---|---|---|---|---|---|---|---|---|---|
| | | Accuracy | F1 | Yes (%) | Accuracy | F1 | Yes (%) | Accuracy | F1 | Yes (%) |
| MiniGPT-4 (13B) | Natural | 70.00 | 72.38 | 58.60 | 62.50 | 67.31 | 64.70 | 63.43 | 68.21 | 65.03 |
| | Synthetic | 66.70 | 71.27 | 56.70 | 58.43 | 66.63 | 64.57 | 57.87 | 66.19 | 64.60 |
| | Δ | **3.30** | **1.11** | **1.90** | **4.07** | **0.68** | **0.13** | **5.56** | **2.02** | **0.43** |
| mPLUG-Owl (7B) | Natural | 60.20 | 70.99 | 87.20 | 53.23 | 67.39 | 93.43 | 53.50 | 67.75 | 94.17 |
| | Synthetic | 58.90 | 70.04 | 87.17 | 52.43 | 67.41 | 53.43 | 52.87 | 67.25 | 93.93 |
| | Δ | **1.30** | **0.95** | **0.03** | **0.80** | -0.02 | **40.00** | **0.63** | **0.50** | **0.24** |
| LLaVA-v1 (7B) | Natural | 62.47 | 72.55 | 86.73 | 55.53 | 69.02 | 93.53 | 53.33 | 68.02 | 95.93 |
| | Synthetic | 60.10 | 71.38 | 83.43 | 53.20 | 68.05 | 92.47 | 52.40 | 67.66 | 93.70 |
| | Δ | **2.37** | **1.17** | **3.30** | **2.33** | **0.97** | **1.06** | **0.93** | **0.36** | **2.23** |
| LLaVA-v1.5 (7B) | Natural | 90.00 | 90.12 | 51.20 | 86.40 | 87.01 | 54.67 | 79.70 | 81.74 | 61.17 |
| | Synthetic | 84.37 | 83.57 | 45.17 | 81.37 | 81.01 | 48.10 | 74.73 | 75.84 | 54.60 |
| | Δ | **5.63** | **6.55** | **6.03** | **5.03** | **6.00** | **6.57** | **4.97** | **5.90** | **6.57** |
| QWen-VL (13B) | Natural | 86.07 | 84.18 | 38.07 | 84.90 | 83.27 | 40.23 | 82.73 | 81.29 | 42.27 |
| | Synthetic | 78.37 | 73.33 | 31.10 | 76.83 | 72.23 | 33.43 | 74.97 | 70.70 | 35.43 |
| | Δ | **7.70** | **10.85** | **6.97** | **8.07** | **11.04** | **6.80** | **7.76** | **10.59** | **6.84** |

existence, quantity and the relation semantics in the scene. Detailed instruction is available in the Appendix. As shown in Figure 3, the length of the revised caption is generally in line with the word limit set by the generative model, ensuring that all key semantic information can effectively prompt the generative model.

## 3.2 Image Synthesis and Filtering

Given the revised caption, we utilize Stable-Diffusion v1.5 to synthesize a set of candidate images. As shown in Figure 2 (right), we apply a filtering strategy to the candidate set, resulting in the final synthetic image with the most authentic and consistent semantics.

**Image Synthesis**: The revised captions are employed as input prompts for image synthesis. We can more easily sample the synthetic image that are similar to natural image by increasing the sampling times. Therefore, we adopt an over-sampling strategy by conducting multiple generations with different random seeds to obtain a set of candidate images.

**Image Filtering**: The filtering process includes two stages: (1) Ensuring authentic semantics in synthetic images, with a focus on avoiding (i) the depiction of objects not existing in natural images or (ii) introducing objects that contradict human cognition. To achieve this, we initially extract objects using automated segmentation tools [35]. Subsequently, we eliminate images displaying an excess or absence of objects in their annotation results when compared to the corresponding natural image. (2) Maintaining consistent semantics with natural images, with a focus on the similarity between synthetic and natural images. Specifically, we filter the candidate set from two dimensions: (i) Image perceptual similarity, referred to as the perceptual system's understanding of the similarity between two images (e.g., high-level semantics in terms of attributes and relations of objects). We use DreamSim [7], which better corresponds to

human perception, to measure the perceptual similarity between synthetic and natural images. (ii) Image semantic faithfulness, referred to as the alignment with textual annotations (e.g., existence-level semantics). Specifically, we compute the cosine similarity between the synthetic image and textual annotations through the CLIP [20] model. Detailed settings are available in the Appendix. Finally, we select the images with lower DreamSim scores and higher CLIP scores as the final synthetic images.

## 4 HALLUCINATIONS ON SYNTHETIC IMAGES

After excluding the influence of synthetic image quality issues on hallucination evaluation, this section delves into the synthetic image-induced hallucination. To more intuitively reflect the impact of synthetic images, we also report the hallucination results on the corresponding natural images and conduct a fair comparison in the context of consistent semantics. Specifically, we first introduce the evaluation datasets and metrics to ensure a comprehensive examination (Section 4.1). Subsequently, we quantify the synthetic image-induced hallucinations across different LVLMs and datasets from perspective of hallucination quantity and position distribution (Section 4.2). Finally, we further investigate the effects on hallucination bias through experiments involving 1) prompt templates, and 2) generation temperatures (Section 4.3).

### 4.1 Experiment Setup

**Dataset**: We selected two widely used datasets, POPE [13] and AMBER [23], as benchmarks for hallucination evaluation. Synthetic images corresponding to these datasets are obtained through semantics translation method. POPE focuses on existence-type hallucination, comprising 500 images with corresponding 9000 annotations. AMBER offers a richer setting with diverse dataset scaling, reasoning

**Table 2: The overall evaluation results of AMBER on both synthetic and natural images. We consider one generative and three discriminative tasks, including the understanding on existence, attribute and relation semantics of objects.**

| Model | Image | EXISTENCE | | ATTRIBUTE | | RELATION | | GENERATIVE | | AMBER |
|---|---|---|---|---|---|---|---|---|---|---|
| | | Accuracy | F1 | Accuracy | F1 | Accuracy | F1 | CHAIR (↓) | Cover (↑) | |
| MiniGPT-4 (7B) | Natural | 10.70 | 19.30 | 54.90 | 39.30 | 57.20 | 29.50 | 22.00 | 59.50 | 51.46 |
| | Synthetic | 10.50 | 19.00 | 54.40 | 39.40 | 57.60 | 27.30 | 23.70 | 52.40 | 50.56 |
| | Δ | **0.20** | **0.30** | **0.50** | -0.10 | -0.40 | **2.20** | **-1.70** | **7.10** | **0.90** |
| MiniGPT-4 (13B) | Natural | 82.80 | 90.50 | 58.80 | 48.10 | 55.80 | 60.30 | 14.80 | 59.80 | 68.62 |
| | Synthetic | 60.20 | 75.10 | 57.40 | 42.80 | 55.50 | 53.40 | 17.20 | 49.20 | 63.46 |
| | Δ | **22.60** | **15.40** | **1.40** | **5.30** | **0.30** | **6.90** | **-2.40** | **10.60** | **5.16** |
| mPLUG-Owl (7B) | Natural | 17.00 | 29.00 | 56.10 | 33.80 | 60.50 | 28.50 | 22.10 | 49.90 | 54.54 |
| | Synthetic | 17.20 | 29.30 | 54.80 | 29.70 | 58.40 | 28.70 | 22.20 | 45.30 | 53.53 |
| | Δ | -0.20 | -0.30 | **1.30** | **4.10** | **2.10** | -0.20 | **-0.10** | **4.60** | **1.01** |
| LLaVA-v1 (7B) | Natural | 16.20 | 27.90 | 66.20 | 57.30 | 52.40 | 61.20 | 11.7 | 49.9 | 69.41 |
| | Synthetic | 5.50 | 10.40 | 61.00 | 48.60 | 61.90 | 56.10 | 14.2 | 47 | 62.84 |
| | Δ | **10.70** | **17.50** | **5.20** | **8.70** | -9.50 | **5.10** | **-2.5** | **2.9** | **6.57** |
| LLaVA-v1.5 (7B) | Natural | 72.80 | 84.20 | 73.20 | 66.00 | 72.70 | 69.50 | 7.20 | 50.80 | 83.44 |
| | Synthetic | 70.90 | 82.90 | 67.80 | 61.40 | 68.60 | 67.90 | 12.10 | 43.60 | 79.17 |
| | Δ | **1.90** | **1.30** | **5.40** | **4.60** | **4.10** | **1.60** | **-4.90** | **7.20** | **4.27** |
| QWen-VL (13B) | Natural | 82.50 | 90.40 | 81.80 | 81.10 | 71.50 | 60.70 | 6.80 | 49.30 | 86.79 |
| | Synthetic | 87.40 | 93.20 | 70.10 | 73.10 | 55.80 | 61.70 | 10.00 | 33.80 | 83.00 |
| | Δ | -4.90 | -2.80 | **11.70** | **8.00** | **15.70** | -1.00 | **-3.20** | **15.50** | **3.79** |

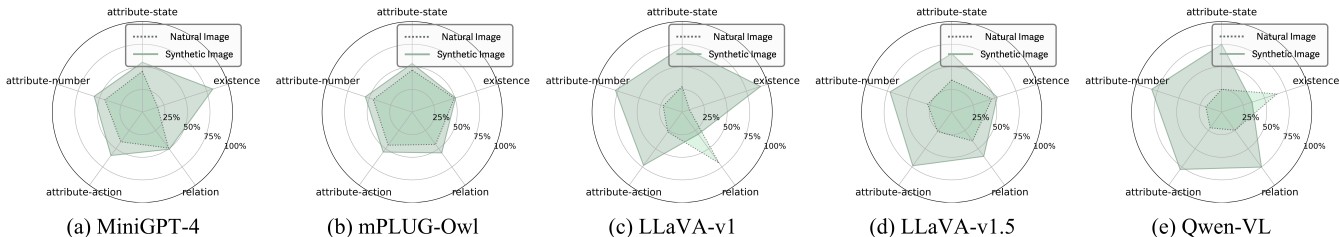

| (a) MiniGPT-4 | (b) mPLUG-Owl | (c) LLaVA-v1 | (d) LLaVA-v1.5 | (e) Qwen-VL |
|---|---|---|---|---|

**Figure 4: Hallucination statistics on different discriminative tasks reasoning within each pair of synthetic and natural image. Discriminative task consider reasoning on attribute, existence and relation semantics, separately. We highlight that the attribute semantic contains the action, number and state information of the annotated objects, separately.**

tasks, and hallucination types. Specifically, AMBER 1) includes 1004 images with corresponding 15200 annotations; 2) assesses both generative and discriminative tasks reasoning and; 3) encompasses three types of model hallucination, including existence, attribute, and relation.

**Metrics on Generative Task Reasoning**: We follow the settings in AMBER, where CHAIR and Cover are used to evaluate the hallucination. CHAIR measures the frequency of hallucinated objects in the responses, while the Cover refers to as the coverage of objects occurring in natural images. Generally, an ideal response should maintain a low hallucination level without sacrificing response quality too much, which means a lower CHAIR and a higher Cover.

**Metrics on Discriminative Task Reasoning**: The hallucination evaluation for the discriminative task is usually defined as a binary

classification. Considering the imbalanced distribution of yes and no answers in the question annotations, we referred to POPE and adopted various metrics, including Accuracy, Precision, Recall, and F1-score. Additionally, we report the 'Yes' ratio in POPE to reveal the confidence behavior of LVLMs.

**Model to be Evaluated**: We conduct hallucination evaluation on the current mainstream LVLMs, including MiniGPT4 (13B) [34], LLaVA-v1 (7B) [16], LLaVA-v1.5 (7B) [15], mPLUG-Owl (7B) [29] and Qwen-VL (13B) [2].

## 4.2 Overall Evaluation Results on Synthetic Image-induced Hallucination

**Observation on Hallucination Quantity**: Table 1 and Table 2 present the evaluation results of the mainstream open-source LVLMs

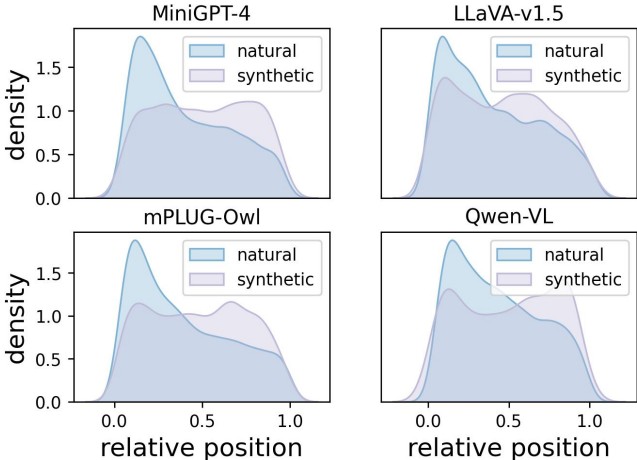

**Figure 5: The relative position distribution of hallucinated objects between synthetic and natural images.**

on both AMBER and POPE datasets. A consistent observation emerges that hallucinations induced by synthetic images are generally more pronounced than those observed in natural images. Additionally, we have the following noteworthy observations: (1) *LVLMs are easily confused by synthetic images in generative tasks reasoning.* This manifests particularly in i) an increased frequency of hallucinated objects and ii) the limited coverage of objects occurring in the image. Moreover, given the consistency constraint on global semantics between natural and synthetic images, it is counter-intuitive for the LVLMs to generate more hallucinations in response to synthetic images. This observation suggests that LVLMs may capture some non-semantic shortcuts beyond the capabilities of the human vision system. (2) *The primary source of synthetic image-induced hallucinations in discriminative tasks reasoning stems from the attribute semantics.* Table 2 provides a detailed comparison among three discriminative tasks, revealing that synthetic images induce higher hallucination results in attribute semantics. Our further analysis includes a comparison of hallucination numbers within each pair of synthetic and natural images across diverse attribute semantics, involving number, action, and state semantics of objects. As shown in Figure 4, synthetic images induce higher hallucinations across all three attributes. (3) *The synthetic image-induced model reasoning behavior appears to be under-confident.* As shown in Table 1, a surprising observation emerges that the 'Yes' ratio in LVLMs reasoning about synthetic images is much lower than that of natural images. This implies that the non-semantic shortcuts in synthetic images weaken the confidence of reasoning process. In other words, synthetic images are more likely to induce the model to say 'No'.

However, on the AMBER result of existence task reasoning, Qwen-VL exhibits higher accuracy on synthetic images, yielding inconsistent behavior. We attribute this disparity to the nature of existence-type hallucination annotations in the AMBER dataset, which consist entirely of counterexample (i.e., questions that consider objects not present in the image). The results in POPE indicate that Qwen-VL has the lowest 'Yes' ratio among all evaluation models, suggesting a low confidence in reasoning discriminative tasks. Given this discrepancy takes advantage of AMBER's annotation to some extent, it does not impact the overall findings.

**Table 3: Hallucination evaluation on generative task under different templates, where brief-desc and detailed-desc refer to as "*Generate a brief/detailed caption of the image*", separately. Red indicates a more severe hallucination bias.**

| Model | Image | brief-desc | | detailed-desc | |
|---|---|---|---|---|---|
| | | CHAIR (↓) | Cover (↑) | CHAIR (↓) | Cover (↑) |
| MiniGPT-4 (13B) | Natural | 5.30 | 32.20 | 13.50 | 58.00 |
| | Synthetic | 7.70 | 28.50 | 16.90 | 48.50 |
| | Δ | **-2.40** | **3.70** | **-3.40** | **9.50** |
| mPLUG-Owl (7B) | Natural | 9.70 | 39.70 | 20.60 | 48.60 |
| | Synthetic | 12.50 | 33.20 | 21.00 | 44.30 |
| | Δ | **-2.80** | **6.50** | **-0.40** | **4.30** |
| LLaVA-v1.5 (7B) | Natural | 2.80 | 36.30 | 6.20 | 49.80 |
| | Synthetic | 6.70 | 32.10 | 10.30 | 43.10 |
| | Δ | **-3.90** | **4.20** | **-4.10** | **6.70** |
| Qwen-VL (13B) | Natural | 6.10 | 30.60 | 6.30 | 46.30 |
| | Synthetic | 12.80 | 21.90 | 14.80 | 32.20 |
| | Δ | **-6.70** | **8.70** | **-8.50** | **14.10** |

**Observation on Hallucination Position Distribution**: We mainly focus on generative task reasoning and perform Kernel Density Estimation (KDE) examining the relative position distribution on both synthetic and natural image-induced hallucinated object. As shown in Figure 5, we observe that in LVLMs' responses to natural images, hallucinated objects tend to appear more at the front of the response, corresponding to the peak of the density curve (i,e., blue distribution) located at the beginning of the relative position. In contrast, in responses to synthetic images, hallucinated objects are relatively uniformly distributed across various locations (i.e., purple distribution). This observation directly indicates that LVLMs usually generate more "security contents" at the end of the response. In contrast, synthetic images exhibit a continuous impact on the extrapolation process of LVLMs, thus resulting in higher hallucination results.

## 4.3 Ablation Study

Previous analyses have demonstrated that the hallucinations induced by synthetic images differ from those of natural images, characterized by a greater quantity and a more uniform distribution. We define the above phenomenon as the synthetic image-induced hallucination bias. In this subsection, we further investigate the effects on hallucination bias, specifically from the perspectives of prompt templates and generation temperature.

**Observation on Prompt Templates**: For generative task, AMBER uses the most concise and commonly used generative prompt, "*Describe this image*" to obtain descriptions of images from LVLMs. Drawing inspiration from POPE's design, we opt "*Generate a brief/ detailed caption of the image*", separately, and explore the influence of prompt templates on the hallucination bias. Table 3 presents the evaluation results of different LVLMs under two prompt templates. A noticeable hallucination bias persists for synthetic images, regardless of whether the prompt template is designed for obtaining detailed or brief descriptions.

Intriguingly, we observe that the long-text generation process appears to amplify the hallucination bias, indicating that the trend of increasing the quantity of hallucinated content in synthetic images

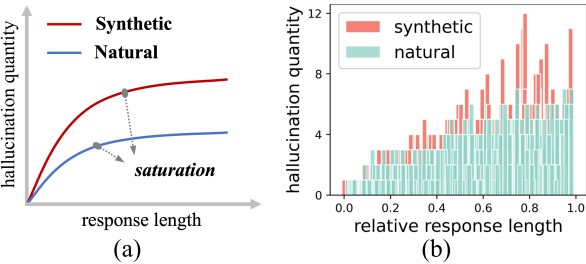

**Figure 6: (a) Quantity evolution of hallucinated objects when response to both synthetic and natural images. (b) The evolution trend of the hallucination quantity with relative response length on MiniGPT-4 (13B).**

surpasses that in natural images. Building upon this finding, we seek insights into the quantity evolution of hallucinated objects in the two types of images. As hypothesized in Figure 6 (a), the hallucination bias in position distribution reveals that synthetic images exert a continuous impact on the extrapolation process, leading to the ongoing generation of hallucinations. In contrast, natural images typically achieve the saturation of hallucinations before synthetic images, thus amplifying the hallucination bias. Nevertheless, the extrapolation length is typically predetermined before the reasoning (e.g., 'max_new_token' generally does not exceed 512), imposing an upper bound on the hallucinations quantity for both two types of images. Results on MiniGPT-4 confirm our hypothesis, as shown in Figure 6 (b).

**Observation on Temperature**: The temperature serves as a crucial hyper-parameter in controlling the randomness and creativity of the text generation process in LVLMs. We investigate the impact on hallucination bias by reasoning under different temperatures. As shown in Figure 7, we observe that synthetic images consistently induce higher hallucinations, regardless of the temperature. Moreover, higher temperatures lead to an increased frequency of hallucinated objects, although it does not noticeably affect the coverage of objects present in the image. This is attributed to the fact that a higher temperature flattens the probability distribution, granting all words an equal chance of selection. Consequently, this increase the probability of generating hallucinated words without significantly altering the generation of objects present in the image.

## 5 INVESTIGATING THE HALLUCINATION BIAS: VIEWPOINT FROM VISUAL PROJECTION MODULE

Previous discussions have confirmed that synthetic images indeed induce LVLMs to generate more hallucinated content. However, synthetic images also exhibit proficiency in successfully engaging in visual segmentation tasks, as evidenced by the consistency of segmentation annotations with natural images (refer to Section 3). The apparent contradiction between these two phenomena raises questions about how synthetic images confuse LVLMs.

It is well-established that an image token is derived by the visual encoding and visual projection processes within LVLMs. Intuitively, the transformation process into an image token may amplify the impact of non-semantic shortcuts. At the same time, given the available difference in the visual representations between natural and

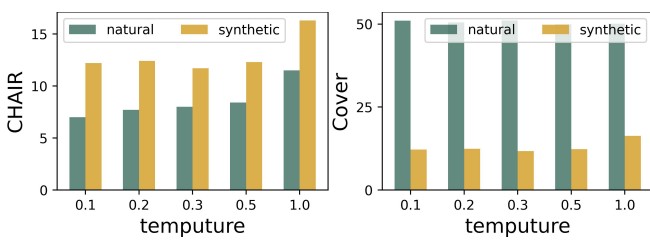

**Figure 7: Hallucination evaluation on generative task under different temperatures, where we report CHAIR and Cover scores on LLaVA-v1.5 (7B) model.**

synthetic images (e.g., Deepfake detection [21] seeks differences between real and fake faces in the feature space.), we are motivated to examine the effects of synthetic image on the visual projection process. Specifically, we analyze two types of LVLMs represented by Q-former (Section 5.1) and Linear (Section 5.2) projection, with a focus on 1) the relative distance between synthetic and natural images in different token spaces and; 2) the correlation with hallucination bias.

### 5.1 Q-former Projection

Q-former, serving as a vision-language connector outfitted with learnable queries for efficient cross-attention mechanisms, stands as a central innovation in BLIP-2 [12]. To investigate the effect of synthetic images on the Q-former projection, in this subsection, we demonstrate the changes in 1) relative distance, and 2) hallucination bias before and after turning off Q-former visual projector.

**Experiment Setup**: We deploy two variants, namely MiniGPT4-Vicuna0 and MiniGPT4-LLaMA-2-Chat, respectively. Both of them utilize the pre-trained BLIP-2 as the vision encoder. Nevertheless, the former retains the Q-former projection in extracting image tokens, while the latter abandons Q-former, aligning with our fundamental requirements. We employ kernel density estimation to visualize the distribution shift of the relative distance between synthetic and natural image tokens. Furthermore, both two LVLMs undergo hallucination evaluation on synthetic images. This exploration aims to uncover the relationship between the Q-former projection and hallucination bias.

**Observations on Relative Distance**: The tokens of synthetic images deviate from those of natural images after Q-former projection. As illustrated in Figure 8 (a), it is observed that the distribution of relative distances significantly decreases when the Q-former is turned off. This suggests that two types of images, which are initially semantically close at the input level, also maintain proximity in the token space.

**Observations on Hallucination Bias**: Synthetic image-induced hallucination bias is amplified by Q-former projection. As depicted in Figure 9 (a), LVLMs without Q-former exhibit a reduced hallucination bias in reasoning for both generative and discriminative tasks compared to preserving Q-former projections. Moreover, this phenomenon aligns with our observation on relative distance. When the Q-former projection is turned-off, the distribution of synthetic images and natural images in the token space gradually converges. Consequently, the issue of hallucinations in synthetic images is rectified to the same level as observed in natural images.

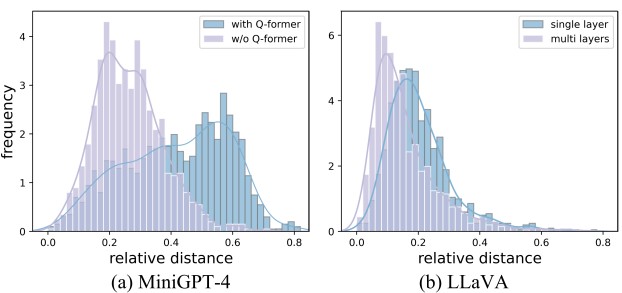

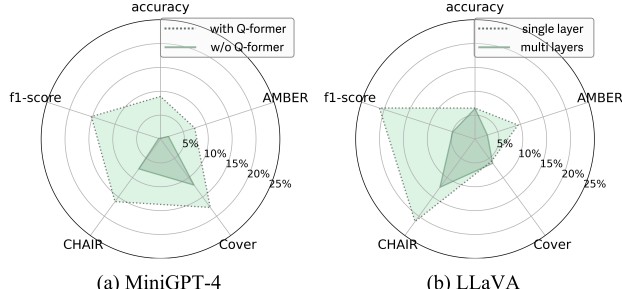

**Figure 8: The changes of relative distance between synthetic and natural images. (a) with or w/o Q-former projection and (b) the number of Linear projection.**

## 5.2 Linear Projection

Linear projection is a significant characteristic of the LVLMs represented by LLaVA architecture, and has emerged as the preferred projection module for many LVLMs owing to its simplicity and effective image-text alignment. Following the same settings on relative distance and hallucination bias mentioned in Section 5.1, this subsection presents two aforementioned observations as we deepen the layers of linear projection.

**Experiment Setup**: We deploy two variants, namely LLaVA-Vicuna and LLaVA-LLaMA2-chat respectively. Both of them used pre-trained CLIP-ViT-Large-34 as vision encoder. Nevertheless, the former utilizes the MLP (i.e., multi-layers linear projection) in extracting image tokens, while the latter retains single layer linear projection.

**Observations on Relative Distance**: The tokens of synthetic images become closer to those of natural images after multi-layers linear projection. Figure 8 (b) reveals a narrowing of the distribution when progressing from a single linear projection to MLP, indicating that deepening the linear projection layer proves effective in reducing the token deviation between synthetic and natural images.

**Observations on Hallucination Bias**: Synthetic image-induced hallucination bias can be reduced by deepening the layers of linear projection. As shown in Figure 9 (b), as the layers of Linear projection decrease, the hallucination bias of the synthetic image increases from the original 4.35 to 6.41. This indicates that the single layer linear projection, fine-tuned with natural text-image pairs, is more susceptible to the influence of synthetic images. Conversely, the MLP structure ensures a closer relative distance between the two types of images, resulting in less hallucination bias for the synthetic image.

## 6 CONCLUSION

Despite the prosperity of generative models, the risks and challenges posed by AIGC cannot be overlooked. This paper pioneers an exploration into the impact of synthetic images on hallucination problems during the reasoning process of LVLMs. Extensive experimental results have confirmed a significant deviation between synthetic image- and natural image-induced hallucination, referring to as the hallucination bias. Further analyses have revealed that the divergence between synthetic and natural image tokens can be attributed to the visual projection module, leading to the amplification of relative

**Figure 9: The changes of hallucination bias after (a) turning off the Q-former projection and (b) deepening the layer of Linear projection. We report the fluctuations in various task reasoning, including discriminative task (refer to as the accuracy and f1-score) and generative task (refer to as CHAIR and Cover) reasoning. We also report the fluctuations on AMBER score.**

distance and hallucination bias. Our future works will endeavor from the following aspects: 1) revealing the cause of synthetic image-induced hallucination bias from the perspective of image synthesis mechanisms and; 2) mitigating the hallucination bias in the reasoning process of LVLMs on synthetic images.

This paper also emphasizes the importance of understanding the differences between natural and synthetic data beyond the applications of forgery detection [21]. As AI-synthetic data becomes more prevalent, we may encounter the following scenarios:

***Scenarios one***: Training with natural data and applying it to natural data. This is the primary focus of current research, where many tasks are well-solved under laboratory conditions. The proposing algorithms achieve performance close to or even beyond human benchmark.

***Scenarios two***: Training with natural data and applying it to synthetic data. This is the scenario discussed in this paper.

***Scenarios three***: Training with synthetic data and applying it to natural data. For instance, the widespread use of the ShareGPT dataset in training large language models, and the potential use of game engine-generated data by Sora. Synthetic data can compensate for the deficiencies of natural data and contribute to continuous improvement in model capabilities. This situation is expected to grow.

***Scenarios four***: Training with synthetic data and applying it to synthetic data.

Both scenarios two and three can be seen as generalized Out-Of-Distribution (O.O.D.) problems, which can be referred as "Natural-Synthetic O.O.D." They address the generalization from natural to synthetic data and vice versa. In fact, even in scenario four, some form of mixing natural and synthetic data for training should be considered. Therefore, understanding the differences between natural and synthetic data is not only essential for authenticity verification but also crucial for better utilization of synthetic data in training and effective interaction with synthetic data in applications.

## ACKNOWLEDGMENTS

This work is supported by the National Key R&D Program of China (No. 2023YFC3310700) and the National Natural Science Foundation of China (No. 62172094).

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
