# OpenReview forum: "AIGCs Confuse AI Too: Investigating and Explaining Synthetic Image-induced Hallucinations in Large Vision-Language Models"
_acmmm.org/ACMMM/2024/Conference — MM2024 Poster_

### Official Review · Reviewer_eoFo · 2024-05-24

**Rating:** 4
**Confidence:** 3

**Summary:**

The paper examines how synthetic images induce hallucinations in AI models, particularly in Large Vision-Language Models (LVLMs). It highlights a significant issue where these models, trained primarily on natural images, exhibit biases and inaccuracies when interpreting synthetic images. The study introduces methods like Semantics Translation to generate and evaluate synthetic images that closely mimic natural images in terms of consistency and authenticity. Extensive experiments reveal that synthetic images tend to induce more frequent and uniformly distributed hallucinations across different LVLMs, a phenomenon termed as 'synthetic image-induced hallucination bias'. The paper also explores modifications in model architectures and image processing techniques to mitigate these biases, particularly focusing on visual projection components like the Q-former and Linear projection modules.

**Strengths:**

**S1: Robust Methodology** The introduction of the semantic translation method is a notable strength, as it ensures that the synthetic images used in experiments maintain high fidelity in terms of visual and semantic consistency with natural images. This methodological rigor allows for more accurate assessments of hallucinations.

**S2: Comprehensive Experiments** The paper conducts extensive experiments to quantify and analyze the hallucination biases, employing multiple datasets and metrics. It not only measures the frequency and distribution of hallucinations but also tests the models under various conditions to generalize findings.

**Limitations:**

**W1: Complexity in Implementation** While the methodologies used are robust, they might be complex for practical implementation outside of research environments. The process of generating and revising synthetic images to match natural images in terms of authenticity and consistency requires significant computational resources and expert knowledge.


**W2: Potential for Overfitting** By tightly controlling the generation and processing of synthetic images to mimic natural images, there's a risk of overfitting the models to the experimental setup, which might not entirely reflect real-world scenarios where synthetic images could be more varied.

**Suitability:**

2

---

### Official Review · Reviewer_S5FJ · 2024-05-24

**Rating:** 4
**Confidence:** 4

**Summary:**

This paper explores the impact of Artificial Intelligence Generated Contents (AIGCs) on Large Vision-Language Models (LVLMs), focusing on the phenomenon of hallucinations induced by synthetic images. The study reveals that synthetic images often cause LVLMs to hallucinate more frequently and uniformly compared to natural images. The paper introduces a semantics translation method to create a controlled environment for evaluating hallucinations and examines how different visual projection modules (Q-former and Linear projection) contribute to these hallucinations. The findings highlight the need for better understanding and mitigating the risks posed by synthetic data on AI models.

**Strengths:**

Novelty: The paper addresses a relatively unexplored area by investigating how synthetic images impact LVLMs, which is crucial given the increasing use of AIGCs in various applications. This focus on AI-to-AI interactions adds a new dimension to the existing research on AI-generated content.

Comprehensive Evaluation: The paper utilizes extensive experiments to compare hallucinations in LVLMs induced by synthetic and natural images. The use of multiple datasets (POPE and AMBER) and various evaluation metrics (e.g., CHAIR, Cover) provides a thorough assessment of the hallucination phenomenon.

Detailed Analysis: The investigation into the effects of different visual projection modules on hallucinations is detailed and insightful. The paper clearly explains how the Q-former and Linear projection modules influence the distribution and quantity of hallucinations, contributing to a deeper understanding of the underlying mechanisms.

Clarity: This paper is well-written and easy-to-follow.

**Limitations:**

Scope of Synthetic Data: While the paper provides valuable insights into the impact of synthetic images, it primarily focuses on specific types of synthetic data generated through semantics translation. The results might not be generalizable to all forms of synthetic data, such as those generated by other models or with different properties.

Potential Overfitting: The modifications proposed for the visual projection modules, such as turning off the Q-former or deepening the Linear projection, may address hallucinations in the specific experimental setups used but might not generalize well to other scenarios or datasets.

**Suitability:**

3

---

### Official Review · Reviewer_dtgs · 2024-05-29

**Rating:** 3
**Confidence:** 3

**Summary:**

This paper examines the impact of AIGCs on Large Vision-Language Models (LVLMs).

**Strengths:**

1. This paper tackles the crucial issue of visual hallucinations in LVLMs, a topic of great significance to the community.
2. The workload of this paper is sufficient.

**Limitations:**

1. The writing of the paper has room for improvement in terms of organization and readability.
2. The claim about the synthetic image quality issues (line 388) is not reasonable. Relying on the filtering process based on similarity scores cannot prove that the quality of the synthetic images is good enough. I think the influence of synthetic image quality issues still exists, resulting in bad evaluation in test period.
3. The overall pipeline is not that reasonable. GPT-4V, as an LVLM, may also exhibit hallucination in the caption period. Additionally, GPT-3.5 ignored the information "there are two trees in this image" and provided an incorrect revision. The content consistency between the natural images and their captions is questionable.
4. The experiment results in Table 1 is not pronounced enough to support the conclusion. Especially considering that synthetic images are not as good as natural images.
5. Minor issues. GPT-4v was incorrectly labeled as GPT4-V in Figure 2.

**Suitability:**

2

---

### Meta-Review · Area_Chair_mWes · 2024-06-26

**Recommendation:** Accept (Poster)
**Confidence:** 3

**Metareview:**

The paper received nearly positive ratings initially. The reviewers think the paper is novel and interesting. The main concerns are some of the claims are not convincing and more comprehensive analysis is needed. After the rebuttal, most of the reviewers' concerns have been addressed. AC made the acceptance decision.